# Cutaneous Applications of the Antiviral Drug Cidofovir: A Review

**DOI:** 10.3390/jcm13092462

**Published:** 2024-04-23

**Authors:** McKayla Poppens, Amanda Ruci, Jeremy Davis

**Affiliations:** 1David Geffen School of Medicine, University of California Los Angeles, 10833 Le Conte Ave, Los Angeles, CA 90404, USA; 2Division of Dermatologic Surgery, Department of Medicine, David Geffen School of Medicine, University of California Los Angeles, 2020 Santa Monica Blvd #510, Los Angeles, CA 90404, USA

**Keywords:** cidofovir, herpes simplex virus, human papillomavirus, molluscum contagiosum, cancer, melanoma, basal cell carcinoma

## Abstract

**Background/Objectives:** Cidofovir, an antiviral drug approved for cytomegalovirus retinitis, has emerged as an alternative treatment option for virally induced cutaneous and mucocutaneous conditions, as well as being trialed as a treatment for select neoplasms. In this review, we highlight the existing evidence, clinical uses, and rationale of using cidofovir for the treatment of cutaneous pathologies. **Methods:** A PubMed database literature search was conducted to identify relevant articles for inclusion in this review. **Results:** Cidofovir has several cutaneous applications in various formulations including intravenous, topical, and subcutaneous administrations. Primarily through case reports, case series, and retrospective reviews, cidofovir has demonstrated efficacy in treating a variety of virally induced conditions—verruca vulgaris, herpes simplex virus, molluscum contagiosum—as well as in adjuvant treatment for select neoplasms. The drug has shown efficacy in immunocompromised and immunocompetent adults and children alike. **Conclusions:** The body of literature supports the use of cidofovir as an effective and well-tolerated treatment for many viral cutaneous pathologies, and encourages further study for its use as an adjuvant therapy for neoplastic disease.

## 1. Introduction

Cidofovir ([S]-1-[3-hydroxy-2-phosphonylmethoxypropyl] cytosine; HPMPC, Vistide) is a nucleoside analog of deoxycytidine monophosphate, originally FDA approved in 1996 for the treatment of cytomegalovirus retinitis in patients with AIDS [1]. It functions via the specific inhibition of viral DNA synthesis by incorporating into a growing DNA strand, acting as a competitive inhibitor and alternative substrate for CMV DNA polymerase, and blocking subsequent viral DNA synthesis [1]. Cidofovir inhibits viral DNA polymerase 35 to 40 times more avidly than human DNA polymerase, rendering it an effective inhibitor of viral replication, while having little to no effect on human cells [2]. Furthermore, human CMV DNA polymerase is unable to remove incorporated cidofovir diphosphate from viral DNA, which may contribute to its prolonged activity beyond the known 17 to 65 h half-life of the active metabolite. In vitro and in vivo, cidofovir has proven effective against virtually all DNA viruses, including herpesviruses [herpes simplex virus (HSV) type 1 and type 2, cytomegalovirus, Epstein–Barr virus (EBV)], and papova-, adeno-, irido, hepadna-, and poxviruses [1]. 

Its antiviral activity against a broad range of DNA viruses has led to its clinical emergence as an off-label treatment for many virally induced pathologies in topical, intravenous (IV), and intralesional (IL) formulations [3]. Multiple case reports and case series support each delivery modality, though no comparison studies of their relative efficacies have been reported. While topical cidofovir has the benefit of the ease of application, IL administration may be preferred due to the precision, improved bioavailability, and, in some cases, shorter duration of treatment [3]. IV administration has been used in cases of extensive disease or critical illness [4,5].

The use of cidofovir for recurrent respiratory papillomatosis is well supported in the literature [6,7,8]. Its use for recalcitrant cutaneous warts has gained traction in recent years, though less is known about its use for other cutaneous pathologies. This review aims to describe the use and effectiveness of all cutaneous applications of cidofovir.

## 2. Purpose

The purpose of this study is to review the clinical use, efficacy, and safety of cidofovir for the treatment of cutaneous pathologies. 

## 3. Materials and Methods

We conducted a literature review using the PubMed database and the search term ‘cidofovir’, both alone and in combination with ‘topical’, ‘intralesional’, and ‘intravenous’, to identify the pool of published articles related to the therapeutic applications of cidofovir. Article titles and abstracts were reviewed for clinical relevance and inclusion in this article. Articles focused on the treatment of non-cutaneous conditions were excluded.

## 4. Pharmacology

The pharmacokinetic properties of cidofovir in humans are available for the IV preparation. Cidofovir exhibits dose-dependent pharmacokinetic properties when systemically administered [9]. Following a single dose, approximately 90% of the drug is recovered in the urine within 24 h [10]. Some data suggest that its elimination from the systemic circulation is a result of active tubular secretion and filtration, as probenecid reduces the renal clearance of cidofovir to that of the creatinine clearance [10]. 

The pharmacokinetic data of topical administrative is available for animal models. The bioavailability of radiolabeled cidofovir on normal, intact rabbit skin was 0.2% to 2.1% and was enhanced by vehicles containing propylene glycol [11]. On abraded skin in a vehicle containing propylene glycol, cidofovir’s bioavailability reached 41%. The topical administration of cidofovir to intact skin lead to negligible systemic drug exposure in the rabbits [11]. In African green monkeys, the pharmacokinetic properties of cidofovir by IV, oral, and subcutaneous routes were measured. Notably, when administered subcutaneously, cidofovir’s bioavailability was 98% ± 15.8% [12]. 

## 5. Dosing and Administration

Topical cidofovir is commonly compounded as a 0.5 to 3% formulation in a cream, ointment, or gel. Daily or twice daily application for at least 1 week and up to 52 weeks has been well tolerated [13]. The practical approaches to IL cidofovir are similar throughout the literature. If multiple injections are required, local anesthesia may be administered to mitigate local pain associated with the IL injection [6]. Studies investigating the use of IL cidofovir for the treatment of all virally induced pathologies have reported diluting cidofovir 75 mg/mL with normal saline to produce concentrations ranging from 2.5 mg/mL to 37.5 mg/mL, with most studies using 15 mg/mL [14,15,16]. IL cidofovir should be injected directly into the lesion using serial puncture or crosshatching techniques [17]. The injected volume of IL cidofovir varies depending on the size and number of lesions, ranging from 0.05 mL for molluscum papules to a maximum of 5 mL for multiple large HSV perineal ulcerations [14,18]. Patients should return on a monthly basis for repeat injections if clinically indicated. 

The literature describing approaches to using IV cidofovir for cutaneous pathologies is limited, though existing case reports follow dosing guidelines similar to its indication for CMV retinitis. The dosages of the successful cases described in the literature ranged from 3.5 to 5 mg/kg [4,5,19,20]. To avoid nephrotoxicity, IV cidofovir was often administered with oral probenecid pre- and post-infusion, as well as IV hydration with normal saline [4,5]. Reported infusion schedules varied from weekly to every other week for up to 20 cycles, based on the clinical response and dependence on renal function [19,20].

## 6. Considerations

While the body of literature regarding cidofovir applications is expanding, large-scale studies are lacking. Nephrotoxicity, neutropenia, and metabolic acidosis are serious side effects of intravenous cidofovir [3]. The reported self-limited side effects of topical applications include inflammation, erosion, or a burning sensation upon application [21]. Although no human studies have been completed to evaluate the bioavailability of topical or intralesional administration, rabbit skin is a reasonable model for comparison. Given this, topical cidofovir should be avoided in the setting of damaged or absent epidermis [22]. 

Several local, self-limiting reactions have been reported regarding intralesional applications, including pain, burning, blistering, discoloration, swelling, and erosion when treating cutaneous warts [15]. Fewer, but similar, localized side effects have been described in non-verrucous applications, including inflammation and ulceration before progressive regression and disappearance [23,24]. All applications of IL cidofovir can expect a similar risk of adverse effects when given a consistent route of administration and dosages. 

With systemic administration, cidofovir was found to be embryotoxic in both rats and rabbits at doses found to be toxic to the mother [1]. In rabbits treated intravenously at 1.0 mg/kg daily, an increased incidence of fetal soft tissue and skeletal anomalies was observed [1]. Because no studies regarding the extent to which topical or subcutaneous applications of cidofovir reaches systemic circulation in humans, or studies concerning the administration of cidofovir via any route to pregnant women are available, cidofovir should be avoided in pregnancy unless the benefit outweighs risk to the fetus [1].

## 7. Clinical Uses

Uses of cidofovir for cutaneous applications have largely been reported in case reports and cohort studies. Despite the limitations of small-scale studies, the literature suggests that cidofovir may be effective in both immunocompetent and immunosuppressed patients [6]. While topical cidofovir has been used more widely, studies have demonstrated that systemic and subcutaneously formulations of cidofovir are safe in patients with HIV/AIDS, lymphomas, leukemias, and those with chronic infection, including disseminated tuberculosis (Table 1) [5,14,18,19,22,25,26,27]. 

### 7.1. Verruca Vulgaris and Verruca Plana

Of all the cutaneous applications of cidofovir, the greatest support exists for its use either topically or intralesionally for verruca vulgaris and verruca plana. While no comparison study exits to determine the relative efficacy of these application methods, many reports in the literature support the use of either, especially in recalcitrant cases or for immunocompromised patients who often present with extensive disease burden. A retrospective analysis of 58 immunocompromised and immunocompetent patients with recalcitrant warts treated with IL cidofovir (15 mg/mL) between 2014 to 2019 found that nearly all patients of both cohorts presented lesional improvement, and more than 75% reached resolution after a mean of 3.4 treatments administered monthly. The locations of verrucae included the hands (74.4%), feet (37.2%), face (7.0%), genitals (9.3%), and other areas of the body (4.7%). All patients had failed several cryotherapy treatments before IL cidofovir was trialed [17].

In 2003, an immunosuppressed kidney transplant patient with generalized verrucae was treated with topical cidofovir 2.5% ointment weekly for 3 weeks, then every other day for an additional 2 weeks [29]. This patient had widespread involvement of her face, bilateral arms, and hands. After completing the 5 weeks of treatment, all lesions significantly improved, though some recurred within a year, prompting a second round of treatment with complete clearance [29]. Another case trialing a course of topical cidofovir 3% cream on a pediatric heart transplant recipient saw the resolution of all facial and digital lesions after 3 months of daily application, and the resolution was sustained at the 12-month follow-up [32]. An immunocompetent pediatric patient with severe verruca vulgaris recalcitrant to many therapies saw near complete resolution after 8 weeks of the daily application of cidofovir 1% cream [33].

IL cidofovir has also been used in pediatric and adult transplant patients with widespread verrucous disease. A pediatric kidney transplant patient who developed acute rejection, CMV, and similar widespread verrucous diseases, was treated with IL cidofovir (7.5 mg/mL), administered at 2 week intervals, which resulted in significant lesion reduction by 6 months. However, the complete resolution of all lesions was only obtained after his immunosuppressive regimen was changed from tacrolimus to rapamycin [29]. In another kidney transplant case with florid palmar disease, after seven sessions of IL cidofovir (7.5 mg/mL) administered in monthly intervals, 95% clearance was achieved without recurrence at 24 months [30]. 

Lastly, in a 2012 retrospective analysis of 12 immunocompetent and immunocompromised children, cidofovir 1% or 3% cream was applied under occlusion either daily or every other day for 5 consecutive days a week, across the course of 3 to 16 weeks. Contrary to other studies, the authors reported less successful outcomes with only 25% of patients reaching clinical resolution and 42% with unchanged lesions, possibly due to the lower concentration in less-frequent dosing (Table 1) [31]. At large, most studies support the use of topical and IL cidofovir for recalcitrant cases, particularly in severely immunocompromised patients; however, more studies are needed to characterize the efficacy, and ti optimize dosing for the treatment of refractory warts.

### 7.2. Verruca Plantaris

Verruca plantaris, commonly referred to as plantar warts, are frequently recalcitrant to traditional therapies, including topical salicylic acid and cryotherapy. As early as 1997, topical cidofovir 3% cream was trialed for a large, treatment-resistant HPV-11, -16, -53, -66 positive verrucous plaque on the foot of a 37-year-old HIV-positive woman [35]. Improvement was noted as soon as 3 days following initiation, with complete clinical resolution by 4 weeks [35]. Since then, several other cases have supported topical cidofovir’s efficacy for recalcitrant plantar warts. From 2008 to 2011, 35 immunocompetent patients with prior unsuccessfully treated plantar lesions were trialed with cidofovir 3% cream twice daily over ~11 weeks. Here, 54.3% of patients experienced complete resolution, while 25.7% had a partial response, and 20% showed no clinical response [36]. Of the patients with complete resolution, only two demonstrated recurrence at roughly the 10-month follow-up [36].

Cidofovir’s efficacy is also supported for plantar warts in immunocompromised children. For twin pediatric patients with severe combined immunodeficiency and extensive plantar involvement, topical cidofovir gel was applied 5 days weekly for a month, followed by water-filtered infrared-A radiation therapy (50–100 J/cm^2^) twice weekly for 5 weeks, which achieved complete resolution by 8 months, and this was sustained at the 2-year follow-up in both patients [39]. Another pediatric sibling pair with widespread planter warts was treated with topical 3% cidofovir under occlusion and saw complete resolution after 5 months [40]. 

Topical cidofovir has even been trialed with laser-assisted delivery. This method of drug delivery was found to enhance the penetration of cidofovir past the epidermis to the dermis using an erbium-doped/yttrium aluminum garnet (Er:YAG) ablative fractional laser prior to the application of topical cidofovir (75 mg/mL). Two patients trialed with this delivery method reported both lesional size and symptom reductions [37]. For one patient, following nine treatments of the laser-assisted delivery of topical cidofovir spaced every 2 to 6 weeks, a 60% improvement in appearance was achieved [37].

The IL delivery of cidofovir has also been used for plantar warts. A patient with a history of lymphoma and a 4 × 5 cm verrucous plaque on the plantar foot was treated with just four treatments of IL cidofovir (75 mg/mL) over a two-month span and had complete resolution at a follow-up four months later (Table 1) [38]. Collectively, these studies provide great evidence to support the use of either topical or intralesional cidofovir for recalcitrant verruca plantaris.

### 7.3. Condyloma Acuminata

The use of cidofovir in recurrent anogenital warts has proven efficacious, especially in immunocompromised patients. In 2002, a study of 74 HIV-positive patients with genital warts were randomized to one of three treatment groups as follows: surgical treatment by electrocautery, topical 1% cidofovir gel for 5 days per week for a maximum of 6 weeks, or electrocautery followed by topical cidofovir 1% gel 5 days per week for 2 weeks [44]. Electrocautery was performed on 29 patients, resulting in a 93.1% lesional clearance; however, 73.7% of the patients in this group experienced reoccurrence at the 6-month follow-up. Topical cidofovir 1% gel was applied to lesions on 26 patients for 6 weeks, yielding a 76.2% complete response, with 35.3% recurrence at the 6-month follow-up. Of the 19 patients who received both electrocautery and cidofovir 1% gel for 2 weeks, all achieved lesional clearance, though 27.3% saw recurrence after the completion of the treatment. Most effective treatments demonstrated by this study included a combination of electrocautery and topical cidofovir [44].

In another case series of twelve HIV-1-positive patients with extensive or relapsed genital warts, three were treated with a combination IL cidofovir (2.5 mg/mL) for 6 weekly administrations, followed by cidofovir 0.5% gel applied every other day until the complete resolution of the lesion was observed [43]. The remaining nine patients were treated with cidofovir 0.5% topical gel following the same timeline. Of the ten patients that were evaluable (three received both IL cidofovir and cidofovir 0.5% gel, seven received only cidofovir 0.5% gel), four patients who received only cidofovir 0.5% gel saw complete resolution within 1–5 applications, and three experienced partial improvement. Of the three patients who received dual therapy, none saw an improvement after six to nine injections of IL cidofovir and two to seven applications of cidofovir 0.5% gel. This study found that newer genital wart lesions were most responsive to cidofovir treatment when compared to older lesions, and that cidofovir should be considered as a reasonable treatment option in HIV-1 positive patients [43]. Furthermore, this study highlights a potential pitfall of using IL cidofovir, given the necessity of several injection points for large lesions. 

In a 9-year-old immunosuppressed pediatric patient with genital warts, cidofovir 1% cream was used following several failed topical therapies [45]. This patient suffered from Fanconi anemia and chronic graft-versus-host disease, and was being concurrently treated with several immunosuppressants (prednisolone, tacrolimus, and mycophenolate mofetil). She developed several papular genital verrucae, to which cidofovir 1% cream was applied daily until complete resolution was achieved after 2 months [45].

IL cidofovir for the treatment of genital warts in solid organ transplant patients has been demonstrated to be safe and efficacious. In an adult male cardiac transplant recipient with perianal and intra-anal warts and with the failed management of surgical excisions and topical imiquimod, single sessions of IL cidofovir (7.5 mg/mL) resolved the lesions within 2 weeks of treatment [29]. To maintain lesional resolution, cidofovir 0.5% topical gel was applied for 3 months. The patient had a recurrence after one year, which was then successfully treated with surgical excision and imiquimod. Another adult transplant patient, a 30-year-old male with a kidney transplant complicated by rejection and on immunosuppression, developed urethral warts and failed treatment with laser and imiquimod before IL cidofovir was initiated. He received four rounds of IL cidofovir (7.5 mg/mL) bi-monthly, which resulted in the resolution of most lesions. The remaining lesions were removed by laser coagulation, and the patient remained lesion free at thew 2-year follow-up (Table 1) [29]. All together, these studies provide evidence of the efficacy of cidofovir in combination with traditional therapies for immunocompromised patients with recalcitrant genital warts (Table 1).

### 7.4. Periungual Verruca

Periungual warts are often challenging to treat, and, many times, are refractory or re-present following conventional treatment. Even more, destructive therapies, including cryotherapy, can cause nail dystrophy. Therefore, cidofovir has been trialed topically and intralesionally in recalcitrant cases. In a chart review from 2010 to 2013, 41 patients who received prior treatment for periungual warts were observed after treatment with topical cidofovir 3% applied once or twice daily [41]. Following this, 56.1% of patients demonstrated the complete resolution of their lesions, and 29.2% had a partial reduction. No response was seen in 14.6% of the patients in this cohort [41]. As part of a larger study using topical cidofovir for plantar warts, 6 of the 11 immunocompetent patients also had periungual involvement treated with topical cidofovir 3% cream twice daily for ~15 weeks [36]. Over 50% of the patients demonstrated complete clinical clearance, roughly 27% experienced a partial response, and 18% had no response. Two of the patients with partial or complete clearance experienced recurrence at the ~9-month follow-up [36]. In another retrospective study of nine patients with recalcitrant periungual warts, IL cidofovir (25 mg/mL) was used and, following an average of 2.7 treatments, all patients experienced improvements in appearance, and seven patients saw the near resolution of the lesions [15]. For three patients, a 95% improvement in lesional appearance was seen after the first IL treatment [15]. 

In a comparison study of sodium tetradecyl sulfate (STS) and IL cidofovir for recalcitrant common warts, 22 patients were treated with IL cidofovir, and, of these, 14 had periungual involvement [42]. IL cidofovir (7.5 mg/mL) was administered monthly until lesional resolution was observed, which was reached after just two sessions for all patients [42]. Furthermore, 26.1% of the control group receiving STS achieved resolution, as compared to the 90.1% of patients treated with IL cidofovir [42].

IV cidofovir has also been trialed for periungual warts in a severe case. A 34-year-old HIV-positive male with periungual warts was successfully treated with IV cidofovir after many failed conventional treatments [20]. The proximal nail and distal interphalangeal joints of multiple fingers were involved. The largest lesion was a 3.5 cm fungating plaque on the right third digit, causing nail deformity. Many prior treatments had been attempted, including cryosurgery, hyperthermia, topical imiquimod, retinols, podophyllin, and 3% topical cidofovir. The intravenous infusion of cidofovir (375 mg) was initiated bimonthly over a total of 5 treatment sessions [20]. Complete resolution was observed after this, except for the residual involvement of the largest plaque. After subsequent surgical resection and an additional two infusions of cidofovir, this lesion completely cleared. This case represents one of the first published cases of the use of IV cidofovir for the successful resolution of localized recalcitrant warts in an HIV-positive patient (Table 1) [20].

### 7.5. Herpes Simplex Virus

Cidofovir has been effectively used for the systemic treatment of mucocutaneous acyclovir- and/or foscarnet-resistant HSV-1 and -2 infections since the 1990s [46]. Because the drug does not rely on viral thymidine kinase for activation, it has demonstrated effectiveness against acyclovir-, ganciclovir- and foscarnet-resistant herpesvirus [3]. 

Following an initial study in 2010, using IL cidofovir in an immunosuppressed patient with HIV suffering from a chronic acyclovir resistant HSV facial ulceration [14], IL cidofovir has been used in several other perineal and palmar HSV lesions. The first patient trialed on IL cidofovir was initially treated with six months of combined IV and IL cidofovir (15 mg/mL), followed by three months of IL-only therapy with significant regression and without evidence of HSV recurrence [14]. Four subsequent patients with extensive genital HSV have been reported in the literature who were treated with two to six rounds of 12.5 to 15 mg/mL IL cidofovir, with near resolution being documented as soon as two weeks following the treatment and resolution as early as six weeks [22,25,26,27]. One patient with a hypertrophic anal lesion was treated with a combination of IL cidofovir and imiquimod 5% cream, and saw no evidence of recurrence three months following treatment [26]. 

Cidofovir has also been trialed topically in immunosuppressed patients for perianal acyclovir- and foscarnet-resistant HSV lesions with mixed efficacy. For one immunosuppressed patient treated with cidofovir 2% ointment twice daily for three weeks, significant wound healing occurred, but the lesion reappeared after therapy discontinuation [47]. In another immunosuppressed patient treated with cidofovir 1% gel, clinical improvement was noted after a week, and the lesion was shown to be HSV-negative through the use of a polymerase chain reaction on repeat testing [48] (Table 2).

### 7.6. Molluscum Contagiosum Virus

Topical cidofovir 3% cream leading to the resolution of recalcitrant M. contagiosum virus lesions in HIV-infected patients was first demonstrated in 1997 [49]. Since then, several case reports and cohort studies have demonstrated the efficacy of cidofovir 1–3% cream for recalcitrant lesions in immunosuppressed adults and children [13,21,34,50,51,53,60,61]. Intravenous administration has been introduced successfully for giant M. contagiosum in immunosuppressed patients [56,57,60]. IL cidofovir recently demonstrated efficacy in a patient with recalcitrant, severe, widespread facial M. contagiosum and coexistent tuberculosis [18]. The patient saw resolution following single 0.05 mL injections into each lesion, and, notably, without adverse reactions or scarring [18]. Both treatment-naïve and -recalcitrant cases of M. contagiosum have been successfully treated with topical and IL cidofovir (Table 2).

### 7.7. Neoplasms

Used as an adjuvant to surgery, IL cidofovir was shown to reduce the number of surgical interventions and achieve the curing of recurrent respiratory papillomatosis [62]. This study quantified the presence of these lymphocytes in conjunction with cell cycle regulators (p16, p53, p63, Ki-67 proteins) in 24 patients with histopathologically confirmed papillomas caused by human papillomavirus (HPV) infection, both before and after treatment with IL cidofovir [62]. The drug influence on the tissue was demonstrated via decreased inflammation (epithelial expression of CD8+) and cell proliferation (p53 and p63) [62]. Although no studies have quantified the drug influence of cidofovir on non-HPV induced tumors, cidofovir’s suppression of cellular proliferation and inflammation may support its antineoplastic properties that have been suggested. 

A study investigating the efficacy of topical 1% cidofovir in four patients with basal cell carcinoma reported tumor regression in histological assessments in three of four patients [54]. Recently, a study using topical cidofovir as a salvage therapy in 23 patients with refractory intra-anal high-grade squamous intraepithelial lesions (HSIL) observed a response in 16 of 23 patients (69.5%), and only non-serious side effects were reported [63]. After a median follow-up of 30.3 months, two of the sixteen patients with a response developed recurrent HSIL (recurrence rate 25.4% at 12 months) [63]. Other studies have suggested thew efficacy of topical cidofovir for HPV-related cervical intraepithelial neoplasia, erythroplasia of Queyrat of the glans penis, lentigo maligna, and intralesionally for Epstein–Barr virus-related nasopharyngeal carcinoma [58,59,64,65].

In a 70-year-old otherwise healthy man with squamous cell carcinoma of the lower eyelid, cidofovir was injected intralesionally and perilesonally once (7.5 mg, 0.1 mL) as an experimental nonsurgical treatment [24]. The patient saw the disappearance of the lesion within a month. A skin punch biopsy of the previous lesional area at 12 months revealed no neoplastic cells, and the patient was free from recurrence at the 24-month follow up [24]. 

Following an article reporting the regression of melanomas in guinea pigs after the systemic administration of cidofovir, a study investigated IL cidofovir for the treatment of cutaneous melanoma metastasis in a single patient [23]. The patient had one pretibial metastatic lesion treated with six rounds of 0.1 mL (7.5 mg) IL cidofovir, and a similar-sized metastatic lesion on the same leg injected with saline for comparison [23]. The lesion treated with IL cidofovir saw disappearance two months after initiating treatment. Both lesions were removed surgically—the one treated with cidofovir revealed an absence of neoplastic cells, while the control showed classical features of intradermal metastasis [23]. 

In 1998, IL cidofovir was trialed in one patient with Kaposi’s Sarcoma, with a series of five weekly injections at a concentration of 2.5 mg/mL [16]. No improvement was reported; however, this study used a significantly lower concentration than has been reported of all other more recent applications of IL cidofovir (Table 2) [16]. 

The accurate assessment of the efficacy of cidofovir as an antineoplastic agent is difficult, given the lack of comparative studies. While some of the studies include histological evidence of clearance, no studies include long-term follow-ups. Additional long-term and comparative studies are needed to better establish the efficacy and safety of cidofovir as a pharmacological agent for mucocutaneous and cutaneous cancers. 

## 8. Discussion

While several virally induced cutaneous pathologies have the potential to spontaneously resolve, persistent and severe cases require therapeutic intervention. Cidofovir has emerged as an efficacious and well-tolerated therapy for recalcitrant cases of many viral pathologies, including human papillomavirus, herpes simplex virus, and poxvirus, and has been used for the treatment of select cutaneous and mucocutaneous cancers. Studies suggest cidofovir is a promising treatment in immunocompetent and immunosuppressed patients alike. Across the literature, cidofovir administered topically was applied for 5 to 7 days each week for an average of 1 to 3 months and for up to a year of daily application. Topical administration is favorable for patients who are unable to tolerate intralesional therapy or traditional destructive methods; however, the duration of the required treatment is a limitation. The literature describing the use of intralesional cidofovir only required single injections in some patients, though most patients were seen back weekly or monthly for up to six rounds of injections. Intralesional application may be preferred for patients who are unable to adhere to a daily topical regimen for several weeks. Further research is needed to delineate the safety and efficacy of cidofovir as well as to compare the efficacies of topical, intralesional, and intravenous administrations. 

Furthermore, while the current systemic administration of cidofovir is available in IV formulation only, its novel lipid conjugate, brincidofovir, has good oral bioavailability and lower rates of nephrotoxicity and myelotoxicity, making it a favorable alternative to cidofovir. Brincidofovir has been developed as a countermeasure to smallpox and is being investigated for the treatment of many other double-stranded DNA viruses, such as cytomegalovirus, adenovirus, human herpes virus-6, BK virus, vaccinia virus, molluscum contagiosum virus, and even cidofovir-resistant varicella zoster virus [66,67,68]. Recently, brincidofovir was used to treat laryngeal HPV16+ squamous cell carcinoma in situ and recurrent respiratory papillomatosis. Of the three patients treated with brincidofovir, two had prior incomplete clearance of their HPV-related laryngeal disease with IL cidofovir (7.5 mg/mL) [69]. As brincidofovir achieves high intracellular concentrations with minimal systemic toxicity, this novel drug may be a promising future oral alternative in cases of widespread viral cutaneous disease and in cases limited by the risk of nephrotoxicity with the systemic administration of cidofovir. 

## Figures and Tables

**Table 1 jcm-13-02462-t001:** Characteristics of patients receiving cidofovir treatment of verruca.

Study	No. of Patients	Age	Sex	Medical History	Failed Treatments	Number and Location of Treated Lesions	Dosing and Administration	Clinical Outcome
Human Papillomavirus—Verruca vulgaris and verruca plana								
Toutous-Trellu et al., 2004 [28]	4	24–50	Male and Female	HIV (CD4 lymphocyte count 5–500 cells/uL; VL 200,000 to <10 copies/mL)	Cryotherapy, electrosurgery, curettage, podophyllin, 5-fluorouracil	3 cases with flat lesions on the forearms, neck, writs, hands; 1 case with raised lesions on the hands and toes	Cidofovir 1% (unk. vehicle) applied daily for 4 weeks	Complete regression in 2 cases of verruca plana, although with recurrence
Kottke et al., 2006 [19]	1	39	Male	HIV-1 (CD4 lymphocyte count 22 cells/uL, VL 58,000 copies/mL), hepatitis B, remote history syphilis, and microsporidiosis	Cryotherapy, 5-fluorouricil cream, salicylic acid solution, imiquimod cream, candida antigen	Disfiguring facial lesions	IV (300 mg) in NS weekly for 2 weeks, then every 2 weeks for 5 cycles	Complete clinical response
Bonatti et al., 2007 [29]	1	32	Male	Renal transplant, autoimmune uveitis, receiving prednisone	None	Generalized lesions on forearms, hands, face	2.5% ointment once weekly for 2 weeks, then every other day for an additional 14 weeks	Recurrence 1-year after treatment; complete clearance after repeating treatment
	1	17	Male	Renal transplant, acute organ rejection	Cryotherapy, surgical resection, imiquimod	Widespread lesions, predominantly face, forearms, hands	Multiple cycles of IL and 2.5% ointment over 6 months	Significant lesional reduction; complete resolution after changing immunosuppressive regimen
	1	34	Male	Bilateral forearm transplant, 3 rejection episodes	None	Multiple wart-like lesions on dorsal fingers and distal hands	1% gel applied every third day; IV also given for CMV infection	Near clinical clearance, persistent lesions stabilized
Cusack et al., 2008 [4]	1	11	Female	Milroy syndrome, asthma	Cryotherapy, cantharidin, salicylic acid, curettage, and diathermy	Bilateral papules on hands and fingers, plantar warts, planar and exophytic lesions on lip and chin	IV (5 mg/kg) weekly for 2 weeks then spaced at 2-week intervals for 5 cycles total	90% resolution after 5 infusions; no recurrence at 10 months follow-up
Blouin et al., 2012 [30]	1	23	Male	Renal transplant patient	Cryotherapy, salicylic acid, cantharidin, podophyllin, imiquimod, bleomycin, CO_2_ laser	More than 100 bilateral exophytic and hyperkeratotic verrucous papules on palms and fingers	7 rounds IL (7.5 mg/mL) at 4-week intervals	95% resolution, no recurrence at 24 months
Gupta et al., 2013 [31]	12	4–16	Male and Female	Immunocompetent and immunocompromised (juvenile rheumatoid arthritis, unspecified immunodeficiency)	Salicylic acid, cryotherapy, imiquimod cream, squaric acid dibutylester, intralesional candida, cimetidine, 5-fluorouracil cream	Feet, hands, knees, face	1% or 3% cream applied under occlusion daily to every-other-day for 5 sequential days/week; average treatment duration, 8.4 weeks	Complete resolution, 25%; partial resolution, 33%; no improvement, 42%
Cleary et al., 2014 [32]	1	8	Female	Heart transplant	Salicylic acid, imiquimod, cantharidin	Multiple lesions on the hands, digits, nose, chin, perioral area	3% cream daily for 2–3 months	Complete resolution of all lesions after 2–3 months; no recurrence at 12 months
Nickles et al., 2019 [33]	1	10	Male	Immunocompetent	Salicylic acid, candida antigen	50+ papules over hands and face, periungual involvement of all fingers of both hands	1% daily for 8 weeks	Near complete resolution without recurrence at 6 months
Alsaleemi et al., 2020 [34]	1	18	Male	Jacobsen syndrome, Paris-Trousseau syndrome	Imiquimod, salicylic acid	Numerous lesions on both hands	1% ointment for 6 months	Complete resolution without recurrence 2 years later
Anshelevich et al., 2021 [17]	58	10–70+	Male and Female	Immunocompetent and immunocompromised (HIV, diabetes, cancer, organ transplant)	Cryotherapy, salicylic acid, duct tape	1–15+ lesions on hands (74.4%), feet (37.2%), face (7.0%), genitals (9.3%), body (4.7%)	IL (15 mg/mL) monthly; average number of sessions, 3.4 ± 2.5	Complete resolution, 75.9%; partial improvement, 98.3%
Human Papillomavirus—Verruca plantaris								
Davis et al., 2000 [35]	1	37	Female	HIV (CD4 lymphocyte count 216 cells/uL, VL < 500)	Clindamycin, itraconazole	Large plaques on medial and lateral right foot	3% cream applied twice daily	Dramatic improvement after 3 days, complete resolution within 3–4 weeks
Padilla Espana et al., 2014 [36]	35	6–55	Male and Female	Immunocompetent	Cryotherapy, keratolytic agents	Plantar (62.9%), hands (31.4%), periungual (17.1%), body (5.7%)	3% cream applied twice daily for ~11 weeks	Complete resolution, 54.3%; partial resolution, 25.7%; no response, 20%; 2 recurrences
Coates et al., 2019 [37]	2	59	Male	Immunocompetent	Excision, salicylic acid, cantharidin	5.5 × 4 cm yellow hyperkeratotic verrucous mass on right posterior heel	9 rounds of 1 mL IL (75 mg/mL) applied under occlusion after ER:YAG laser spaced every 2–6 weeks	~60% reduction in lesional size, islands of complete clearance, markedly reduced hyperkeratosis
Moore et al., 2015 [38]	1	Unk.	Unk.	Lymphoma, lenalidomide	Cryotherapy, electrocautery, salicylic acid, intralesional bleomycin, laser	4 × 5 cm hyperkeratotic white plaque on right foot	4 rounds IL (15–25 mg/mL)	Complete resolution sustained at 4-month follow-up
Marini et al., 2006 [39]	2	9	Female	SCID	Surgical excision, topical viricidal agents, topical imiquimod 5% cream	Multiple confluent, exo- and endophytic lesions on plantar feet and toes	1.5% gel applied 5 days/week for 4 weeks followed by irradiation twice weekly for 5 weeks	Complete resolution without recurrence
Henrickson et al., 2017 [40]	2	7, 12	Male	SCID	Liquid nitrogen, imiquimod, salicylic acid, sinecatechins urea 40% cream, 5-fluorourcil	Bilateral plantar lesions	3% cream under occlusion for 5 months	Complete resolution after 5–6 months
Human Papillomavirus—Periungual verruca								
Hivnor et al., 2004 [20]	1	34	Male	HIV (CD4 lymphocyte count 21 cells/uL)	Cryosurgery, hyperthermia, imiquimod, tretinoin solution, tazarotene gel, podophyllin	Multiple enlarging papules on proximal nail fold and distal interphalangeal join of right third finger	3% ointment followed by IV (375 mg) every 2 weeks for 5 sessions	Near complete resolution after 5 IV sessions; residual lesion on 3rd digit resolved with surgical excision and 7 IV sessions
Grone et al., 2006 [5]	1	21	Female	Myelodysplastic syndrome, receiving chemotherapy and stem cell factor therapy	None	Enlarging lesions on upper trunk, shoulders, hands affecting lateral nail folds	IV (375 mg [3.5 mg/kg]) weekly for 2 weeks, then twice monthly for 18 weeks	Reduction of all lesions, persistent lesions on hand treated with CO_2_ laser; new lesions at 1-year follow-up
Padilla Espana et al., 2014 [41]	41	4–60	Male and Female	Immunocompetent	Cryotherapy, electrocautery, laser, photodynamic therapy, trichloroacetic acid, other keratolytics	Several lesion locations involving the extremities and body	3% cream applied once or twice daily	Complete resolution, 56.1%, partial response, 29.2%, no response, 14.6%; 3 recurrences at follow up
Oh. 2020 [42]	14	12–65	Male and Female	Immunocompetent	Cryotherapy, bleomycin intralesional injection	Multiple periungual, dorsal and palmar hand lesions	1–3 rounds IL (7.5 mg/mL) monthly	Clinical resolution, 100%; 9% recurrence at 2-month follow-up
Poppens et al., 2023 [15]	9	20–89	Male and Female	Immunocompetent and immunocompromised	Cryotherapy, candida, electrodessication, 5-flurouracil, imiquimod, salicylic acid	Peri- and subungual lesions	Mean of 2.7 rounds of IL (25 mg/mL)	Complete or near complete resolution, 77.8%; clinical improvement, 100%
Human Papillomavirus—Condyloma acuminata								
Orlando et al., 1999 [43]	12	Unk.	Unk.	HIV-1 (Mean CD4 lymphocyte count 318.3 cells/uL, VL 361 copies/mL)	Surgical excision, intralesional probenecid	Anogenital lesions	2 groups: IL followed by 0.5% gel or only 0.5% gel	Dual therapy, no response; topical group, 50% complete resolution, 50% partial resolution
Orlando et al., 2002 [44]	74	24–41	Male and Female	HIV (Mean CD4 lymphocyte count 264.9 cells/uL, mean VL 34.6 copies/mL)	None	Penial, vulvar, perineal, perianal	3 treatment groups: electrocautery, 1% gel (5 days/week for 6 weeks), combined therapy for 2 weeks	Electrocautery only, 93.1% complete response, 73.7% relapse; cidofovir only, 76.2% complete response, 35.3% relapse; combination therapy, 100% complete response, 27.3% relapse
Bonatti et al., 2007 [29]	1	32	Male	Cardiac transplant	Surgical removal, imiquimod	Perianal and intra-anal lesions	IL (7.5 mg/mL) once followed by 3 months gel	Lesion free until 12-month follow-up
	1	30	Male	Renal transplant	Radical laser surgery, imiquimod	Urethral lesions	4 rounds IL (7.5 mg/mL) every 2 weeks	Near clinical resolution, residual lesions removed via laser; no recurrence at 24-month follow-up
Muffarrej et al., 2019 [45]	1	9	Female	Fanconi anemia, stem cell transplant graft vs. host disease	Fusidic acid, aqueous emulsifying ointment, hydrocortisone, miconazole	Multiple papular, cauliflower-like growths in the genital area	1% cream daily for 2 months	Significant improvement after 7 days; complete resolution at 2 months without recurrence at 6-month follow-up

AIDS, acquired immunodeficiency syndrome; HIV, human immunodeficiency virus; INF, interferon; IL, intralesional; IV, intravenous; SCID, severe combined immunodeficiency; Unk., unknown.

**Table 2 jcm-13-02462-t002:** Characteristics of patients receiving cidofovir treatment of non-verrucous pathologies.

Study	No. of Patients	Age	Sex	Medical History	Failed Treatments	Number and Location of Treated Lesions	Dosing and Administration	Clinical Outcome
Herpes Simplex Virus								
Snoeck et al., 1994 [46]	1	41	Male	HIV/AIDS, bone marrow transplant	Oral acyclovir, IV foscarnet, gancyclovir	Perianal and anal lesions	1% cream for 3–4 days for 8 cycles	Regression of most lesions, recurrent lesions treated with acyclovir
	1	37	Male	CML	Oral acyclovir, IV foscarnet	Buccal and facial lesions	1% cream for 3 days	Facial lesions resolved, residual buccal lesion treated with acyclovir
Castelo-Soccio et al., 2010 [14]	1	34	Male	HIV (CD4 lymphocyte count 400 cells/uL, VL undetectable)	IL steroids, oral, IV acyclovir, vancomycin, surgical debridement	1 large exophytic nasal lesion with ulceration	6 months of combined IV and IL therapy, followed by 3 months of only IL (75 mg/mL, diluted 1:4 with saline increased to 75 mg/mL, diluted 1:1 with saline)	Significant improvement when IL was started and continued; significant lesion regression and no HSV lesion recurrence
Evans et al., 2011 [47]	1	60	Male	MCL, allogenic stem cell transplant	Oral valacyclovir, IV foscarnet	Circumferential perianal ulcer	2% ointment applied twice daily for 3 weeks	Significant wound healing until treatment ceased
Muluneh et al., 2013 [48]	1	34	Female	CML	Oral valacyclovir, IV acyclovir, IV foscarnet	3 cm circumferential perianal ulcer	1% gel every 6 h with continuation of IV foscarnet	Improvement after 1 week and HSV negative using PCR
Wanat et al., 2013 [22]	1	55	Male	HIV (CD4 350 cells/uL, VL undetectable), hepatitis C virus	Oral acyclovir, valacyclovir, famciclovir	Multiple exophytic, verrucous and ulcerated plaques on scrotum and perianal area	3 doses IV followed by 6 rounds IL spaced every other week (1:4 dilution of 75 mg/mL, 5 mL into scrotal lesion and 5 mL into perianal lesion)	Improvement after 3 doses IV (discontinued after rising serum creatinine), resolution of scrotal lesion and dramatic improvement in perianal lesion after 6 IL treatments
Nieto Rodriguez et al., 2017 [26]	1	64	Female	Follicular Hodgkin lymphoma, bone-marrow transplant	Oral acyclovir	Hypertrophic anal lesion	Single injection (0.3 mL, concentration unspecified) followed by topical imiquimod 5% cream 3-times weekly	Complete resolution and lack of recurrence at 3 months
Saling et al., 2019 [27]	1	51	Female	AIDS (CD4 lymphocyte count 184 cells/mm, VL < 20 copies/mL) prior herpetic whitlow with underlying osteomyelitis	Amoxicillin-clavulanic acid, famciclovir, imiquimod cream	1 raised, hypertrophic lesion on palmar right hand	3 rounds IL (12.5 mg/cc, 0.2–0.4 mL)	Complete resolution 6-weeks following last dose; no recurrence 3 years later
Enescu et al., 2021 [25]	1	69	Male	HIV/AIDS (CD4 lymphocyte count 9 cells/mm), extensive recurrent genital and oral HSV, scrotal and urethral meatus SCCIS, ESRD	Imiquimod 5% cream, oral valacyclovir	3 ulcerated plaques on left inguinal crease, right medial thigh, and penis	2 rounds IL spaced 2 weeks apart (15 mg/mL)	Near resolution 2 weeks after second treatment
Molluscum Contagiosum Virus								
Meadows et al., 1997 [49]	1	26	Male	HIV (CD4 lymphocyte count 90 cells/uL; VL < 500 copies/mL)	IV ganciclovir, electrodessication and curettage, INF-alpha, PDL	Nodular confluent plaques on the face, widespread lesions on chest, arms, perianal area	initiated for CMV retinitis: IV (2 mg/kg) weekly for 2 weeks, then every 2 weeks for 2+ months	Facial and truncal lesions resolved by 2 months; patient remained on maintenance regimen without recurrence
	1	37	Male	HIV (CD4 lymphocyte count 210 cells/uL)	Cryotherapy, tretinoin, podophyllin	Umbilicated plaques on >95% of face	3% cream daily for 1 month	70% lesions resolved by 2 weeks, all by 1 month
	1	31	Male	HIV (CD4 lymphocyte count 280 cells/uL)	Cryotherapy	Lesions on >90% face and neck	initiated for CMV retinitis: IV (5 mg/kg) weekly for 2 weeks then once every 2 weeks	Complete resolution after 1 month without residual inflammation, remained on therapy for CMV retinitis
Davies et al., 1999 [50]	1	12	Male	Wiskott-Aldrich syndrome	INF-alpha, INF-gamma	Rapidly spreading lesions affecting > 75% skin surface	1% cream applied for 2–3 weeks	Dramatic resolution of all lesions
Zabawski et al., 1999 [51]	1	8	Female	Immunocompetent	None	Scattered umbilicated papules on trunk and upper legs	3% cream applied twice daily for 6 weeks	Complete clearance, no recurrence at 6 months
	1	7	Male	Immunocompetent	None	Eruption of papules on upper chest, arms, axillae, upper legs	3% cream applied twice daily for 4 weeks	65% reduction by 2 weeks, complete clearance at 4 weeks; localized autosensitivity secondary to irritation developed at 2 weeks and cleared with triamcinolone 0.1% ointment
Calista et al., 2000 [21]	4	31–56	Male	HIV/AIDS	Curettage, cryotherapy	Lesions on face, neck	1% cream twice daily for 2–6 weeks	Complete clearance, 75% relapsed 1-month post-treatment
Ibarra et al., 2000 [52]	1	32	Male	HIV/AIDS (CD4 lymphocyte count 76 cells/uL, VL 427,764 copies/mL), pulmonary tuberculosis	Curettage, podophyllin	Umbilicated papules and plaques on face	IV (5 mg/kg) weekly then every 2 weeks for 9 cycles	Complete remission
Toro et al., 2000 [53]	2	4, 8	Male	HIV-1 (CD4 lymphocyte count 168–329 cells/uL; VL 430,000+ copies/mL)	Cryotherapy, cantharidin, tretinoin gel	Hundreds of umbilicated and pearly papules over entire body, including face and perineal area	3% combination vehicle (Dermovate) once daily for 5 days/week over 8 weeks; nonfacial lesions under occlusion	Complete clinical resolution, no recurrence at 18 months
Baxter et al., 2004 [13]	1	29	Male	HIV (CD4 count < 250, VL > 55,000 copies/mL), M. avium intracellulare infection, diabetes, neutropenia, transfusion dependent anemia	Cryotherapy, curettage and diathermy, podophyllotoxin, diphencyprone contact sensitization	Giant necrotic suppurating facial lesions	3% combination vehicle (Dermovate) applied once daily for 5 days/week for 12 months; IV tried but discontinued (renal dysfunction)	70% improvement after 3 months, no recurrence 2 years post-treatment
Fery-Blanco et al., 2007 [54]	1	35	Male	Atopic dermatitis	None	Large eruption in zones previously treated with substantial amount of tacrolimus	3% lotion (unk. dosing)	Complete clearance without recurrence
Hicks et al., 2008 [55]	1	43	Male	Wiskott-Aldrich syndrome, recent chemotherapy	Cryotherapy, imiquimod, cantharidin, curettage, topical emollients, salicylic acid, TCA peel	Lesions over entire trunk and lower extremities	Topical (unk. dosing or vehicle)	Treatment halted after left arm edema developed; lesions persisted
Erickson et al., 2011 [56]	1	43	Male	HIV (CD4 lymphocyte count 32 cells/uL)	None	Eruptive nodules on scalp, face, neck,	IV (5 mg/kg) weekly for 2 weeks followed by infusions every other week, 10 doses in total	Dramatic improvement after 5 doses, no new or recurrent lesions at 2-month follow-up; course complicated by secondary MRSA/E. coli infection at 2 weeks
Foissac et al., 2014 [57]	1	22	Male	Post-traumatic splenectomy, acute lymphoblastic leukemia, pneumocystosis, smoking	Curettage, cryotherapy, imiquimod	~50 papules on face, scalp, trunk, limbs, pubis	IV (5 mg/kg) spaced 8 days apart for 7 cycles	Complete remission
Alsaleemi et al., 2020 [34]	1	12	Female	Jacobsen syndrome	Cantharidin	Crops of lesions in the popliteal fossae	1% ointment for 3 months	Complete clearance without recurrence at 3-year follow-up
Quintana-Castanedo et al., 2021 [18]	1	30	Female	HIV/AIDS (CD4 30 cells/uL, VL 1,540,000 copies/mL), disseminated tuberculosis	Topical keratolytics, cryotherapy	Numerous lesions over periocular areas and eyebrows	Single round IL (1% solution, 0.05 mL/lesion)	Complete resolution and no new lesions at 6-month follow-up
Neoplasms								
Simonart et al., 1998 [16]	1	75	Male	Steroid-dependent asthmatic bronchitis	None	1 macule (early stage), 1 papule and 1 nodule (late-stage) of Kaposi’s sarcoma	5 rounds IL spaced weekly (2.5 mg/mL)	No effect
Calista et al., 2002 [24]	1	70	Male	Multiple actinic keratosis	Refused conventional surgery	Cutaneous SCC of lower right eyelid	Single round of IL and PL injection (7.5 mg, 0.1 mL)	Resolution with good cosmetic outcomes and no recurrence at 24 months; punch biopsy at 12 months post injection revealed no neoplastic cells
Calista. 2002 [58]	1	37	Male	Condylomata acuminata	Cryotherapy, CO_2_ laser	Erythroplasia of Queyrat of glans penis	1% cream applied once daily for 5 consecutive days for 2 weeks	Punch biopsy following treatment revealed no neoplastic cells and in situ hybridization of HPV 16/18 was negative
Calista. 2003 [23]	1	45	Male	Subungual melanoma on left hallux	None	1 5 mm cutaneous melanoma metastasis on left leg	6 rounds IL in 2-week intervals (7.5 mg, 0.1 mL)	No presence of neoplastic cells on histologic evaluation following surgical removal
Calista. 2007 [59]	2	75, 80	Female	Lentigo maligna	Surgery	Multiple 1–3 mm lesions on left cheek; 45 × 35 mm macular lesion on right cheek	1% cream applied once a day for 10 days, then every other day for 3 months	Resolution with good cosmetic outcome and no recurrence after 4 years; punch biopsy at 3 months post treatment revealed no neoplastic cells

AIDS, acquired immunodeficiency syndrome; CML, chronic myeloid leukemia; HIV, human immunodeficiency virus; INF, interferon; IL, intralesional; IV, intravenous; MCL, mantle cell lymphoma; PCR, polymerase chain reaction; PDL, pulse-dye laser; PL, perilesional; SCC, squamous cell carcinoma; TCA, trichloroacetic acid; Unk., unknown.

## Data Availability

All data referenced in this manuscript are publicly available.

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
