# Peer review of "Cutaneous Applications of the Antiviral Drug Cidofovir: A Review"

_jcm, 2024, doi:10.3390/jcm13092462_

Round 1
Reviewer 1 Report
Comments and Suggestions for Authors
It is a well-written review article that gives interesting information to the clinician about the use of cidofovir as an effective and well-tolerated treatment in common viral cutaneous disorders.
A few mistakes to be corrected :
1. Section: Abstract, Line 18: please, replace “human” with “herpes”
2. Section: Periungual verruca, Line: 285: please, remove “demonstrates"
Author Response
Dear reviewer, thank you for your time in reviewing our article and for your comments designed to improve our paper. Please find our responses to your comments below.
Abstract
- Line 18: please replace “human” with “herpes”
- Line 18, replaced “human” with “herpes”
Periungual verrucae
- Line 285: please remove “demonstrates”
- Line 285, removed “demonstrates”
Reviewer 2 Report
Comments and Suggestions for Authors
Dear Authors,
it was a great effort to sum up the evidence of a single drug in experimental and clinical setting. Cidofovir seems to be an interesting agent in dermatologic skin infectious virus related.
I have some comments:
- in intro can you add a sentence on the approval status (EMA or FDA) of the drug?
- line 150: "after 3 months of treatment" means once daily/week? Please specify the dosage
- In most of trials/studies included in this review, the efficacy was obtained after months of administration, and this could be a limitation. Thus, its use should be limited to immunocompromised patients or second/third line of therapy. What's your thought about the duration of therapy? Please add a sentence in discussion.
Author Response
Dear reviewer, thank you for your time in reviewing our article and for your comments addressed to improve our paper. Please find our responses to your comments below.
- in intro can you add a sentence on the approval status (EMA or FDA) of the drug?
- Added “FDA” to line 29. The statement about its approval referred to its FDA approval. The statement is now clear with the addition of “FDA”.
- line 150: "after 3 months of treatment" means once daily/week? Please specify the dosage
- Added “daily application” to line 151 for clarification.
- In most of trials/studies included in this review, the efficacy was obtained after months of administration, and this could be a limitation. Thus, its use should be limited to immunocompromised patients or second/third line of therapy. What's your thought about the duration of therapy? Please add a sentence in discussion.
- Added to the discussion in lines 377 to 384 to expand on this point. The following was added: “Across the literature, cidofovir administered topically was applied 5 to 7 days each week for an average of 1 to 3 months and up to a year of daily application. Topical administration is favorable for patients unable to tolerate intralesional therapy or traditional destructive methods, however, the duration of required treatment is a limitation. The literature describing the use of intralesional cidofovir only required single injections in some patients, though most patients were seen back weekly to monthly for up to 6 rounds of injections. Intralesional application may be preferred for patients unable to adhere to a daily topical regimen for several weeks.”